# The Evolving Role of Stereotactic Body Radiation Therapy for Head and Neck Cancer: Where Do We Stand?

**DOI:** 10.3390/cancers15205010

**Published:** 2023-10-16

**Authors:** Issa Mohamad, Irene Karam, Ahmed El-Sehemy, Ibrahim Abu-Gheida, Akram Al-Ibraheem, Hossam AL-Assaf, Mohammed Aldehaim, Majed Alghamdi, Ibrahim Alotain, May Ashour, Ahmad Bushehri, Mostafa ElHaddad, Ali Hosni

**Affiliations:** 1Department of Radiation Oncology, King Hussein Cancer Center, Amman 11941, Jordan; imohamad@khcc.jo; 2Department of Radiation Oncology, Odette Cancer Centre, University of Toronto, Toronto, ON M4N3M5, Canada; irene.karam@sunnybrook.ca; 3Faculty of Medicine, University of Toronto, Toronto, ON M5S1A1, Canada; ahmed.elsehemy@mail.utoronto.ca; 4Department of Radiation Oncology, Burjeel Medical City, Abu Dhabi 7400, United Arab Emirates; ibrahim.abugheida@burjeelmedicalcity.com; 5Emirates Oncology Society, Dubai 2299, United Arab Emirates; 6Department of Nuclear Medicine, King Hussein Cancer Center, Amman 11941, Jordan; aibraheem@khcc.jo; 7Department of Radiation Oncology, Comprehensive Cancer Center, King Fahad Medical City, Riyadh 11525, Saudi Arabia; 8Department of Radiation Oncology, King Faisal Specialist Hospital and Research Center Riyadh, Riyadh 11211, Saudi Arabia; maldehaim@kfshrc.edu.sa; 9Radiation Oncology, Princess Noorah Oncology Center, King Abdulaziz Medical City, Ministry of National Guard Health Affairs-Western Region, Jeddah 21556, Saudi Arabia; alghamdima12@mngha.med.sa; 10College of Medicine, King Saud Bin Abdulaziz University for Health Sciences, Jeddah 11481, Saudi Arabia; 11Department of Radiation Oncology, King Fahad Specialist, Dammam 31444, Saudi Arabia; Ibrahimalotain@moh.gov.sa; 12Department of Radiation Oncology, National Cancer Institute, Cairo University, Cairo 11796, Egypt; may.ashour@nci.cu.edu.eg; 13Department of Radiation Oncology, Kuwait Cancer Control Center, Kuwait 42262, Kuwait; abushehri@moh.gov.kw; 14Clinical Oncology Department, Kasr Al-Ainy Center of Clinical Oncology and Nuclear Medicine, Kasr Al-Ainy School of Medicine, Cairo University, Cairo 12613, Egypt; 15Radiation Medicine Program, Princess Margaret Cancer Centre, University Health Network, University of Toronto, Toronto, ON M5G2M9, Canada

**Keywords:** head and neck cancer, SBRT, hypofractionation

## Abstract

**Simple Summary:**

Currently, Stereotactic Body Radiation Therapy (SBRT) is reserved for head and neck cancer (HNC) patients who are not suitable candidates for conventional radiation therapy and should not be considered as a first line of treatment option and as a boost. It should be performed in the context of clinical trial. This review aims to explore SBRT’s role in different HNC scenarios. It has the potential to greatly impact the clinical practice by providing valuable insights into the appropriate indications for SBRT in HNC treatment; as well as the practical and technical considerations involved in administering SBRT for HNC; SBRT dosage for various HNC scenarios; and treatment results. However, further research is needed to fully investigate these applications.

**Abstract:**

Stereotactic body radiation therapy (SBRT) is a precise and conformal radiation therapy (RT) that aims to deliver a high dose of radiation to the tumor whilst sparing surrounding normal tissue, making it an attractive option for head and neck cancer (HNC) patients who are not suitable for the traditional long course of RT with comprehensive RT target volume. Definitive SBRT for HNC has been investigated in different settings, including early stage glottis cancer, and as an alternative to brachytherapy boost after external beam RT. It is also used as a primary treatment option for elderly or medically unfit patients. More recently, an SBRT combination with immunotherapy in the neoadjuvant setting for HNC showed promising results. Salvage or adjuvant SBRT for HNC can be used in appropriately selected cases. Future studies are warranted to determine the optimum dose and fractionation schedules in any of these indications.

## 1. Introduction

Head and neck cancers (HNCs) constitute about 6% of global malignancies, with approximately 650,000 new cases and 350,000 annual deaths [1]. They often originate from different anatomical sub-sites in the head and neck (HN) region [1], primarily being a squamous cell carcinoma (SCC) [2]. Second primary HNC occurs at rate of 3–5% [3]. HNCs are increasingly prevalent, especially in men, typically diagnosed in their early 60s [4,5,6,7,8]. Treatment options generally include surgery, radiation therapy (RT), systemic therapy, or a combination of any of these according to the overall stage and type of cancer, preference and medical/general condition of the patient, and the intent of treatment [9,10,11]. RT or chemoradiotherapy (CRT) is routinely used in the majority of advanced HNC, lasting usually for 6–7 weeks, as a primary or post-operative therapy [6]. However, some patients cannot tolerate a prolonged RT/CRT course due to their age, comorbidities, travel challenges, or lack of social support [12].

Stereotactic body radiation therapy (SBRT) is a precise HN treatment targeting specific areas with high-doses of radiation delivered in 1 to 5 fractions of ≥5 Gy per fraction using image guidance [12,13,14,15,16,17]. It destroys tumor blood vessels, leading to endothelial cell death [18]. New evidence indicates that SBRT maintains radiation-induced cellular death pathways and possibly enhances anti-tumor immunity with high fractional doses [19]. 

The utilization of SBRT in real-world practice varies between 0 and 10% [12,20,21,22,23,24]. SBRT is increasingly being used in treating a variety of cancers. However, the SBRT indications for HNC, dose, fractionation schedules, and HN organs-at-risk (OARs) dose constraints lack uniform consensus [25]. The data regarding oncologic and toxicity outcomes associated with SBRT for HNC are sparse [21,22,23]. This review aims to summarize the literature for SBRT to HNC in the definitive, neoadjuvant, salvage, and adjuvant settings from clinical and technical perspectives.

## 2. Radiobiological Principles of SBRT for HNC

High-dose radiation per fraction induces more necroptosis and apoptosis. Consequently, the repair of tumor cells becomes almost impossible, or occurs at an exceedingly low rate, leading to the majority of tumor cells suffering from radiation-induced damage. Moreover, a single high-dose SBRT treatment completely halts the cell cycle at all stages, thereby preventing the redistribution of tumor cells. This high-dose radiation effectively eliminates both oxygenated and hypoxic cells, efficiently eradicating the tumor. In contrast, following conventional radiation therapy, accelerated repopulation of tumor stem cells often occurs after approximately three weeks. However, SBRT treatment is typically completed within one week, effectively sparing tumor cells from accelerated repopulation. On the other hand, head and neck squamous cell carcinoma (HNSCC) has a low repair capacity where hyperfractionation can potentially result in better outcomes. In addition, incomplete repair can be a problem for some late-responding normal tissues if large doses are administered without enough interfraction time to allow for a complete repair of sublethal damage. If a significant amount of residual unrepaired damage remains after a too short interfraction time, the accumulation of residual damage to the damage produced by the subsequent SBRT fraction can result in an excess of toxicity to normal tissues. Notably, normal mucosa has a very high repopulation capacity that cannot protect mucosa during SBRT, so it is ideal to keep normal tissues/mucosa out of the high dose volume during SBRT for HNC. Moreover, if the tumor is hypoxic, reoxygenation of the hypoxic region could be possible with a more protracted course rather than a short course SBRT [12,13,14,15,16,17,18]. 

## 3. Practical and Technical Aspects of SBRT for HNC

Compared with SBRT, Intensity modulated radiation therapy (IMRT) is typically administered over a longer course of treatment, often several weeks, and is better suited for larger or more complex tumors. While both techniques aim to deliver effective radiation therapy with minimal damage to healthy tissue, SBRT’s emphasis on precision, accuracy, rapid treatment, meticulous target volume delineation, no or minimal clinical target volume (CTV), possibly tighter planning target volume (PTV) margin, steep dose gradient, and larger dose per fraction make it particularly well-suited for certain clinical scenarios [11,15].

### 3.1. Target Volume Definition for SBRT

The majority of institutions use a cut off size and/or volume constraint for a primary tumor (e.g., 3–5 cm/25–30 cc) and nodal disease (4–5 cm/ <50 cc) [24]. Contouring protocols varied across studies with different approaches taken. At the time of simulation, the use of an intravenous contrast (whenever possible) and magnetic resonance imaging (MRI) diagnostic or simulation scans (whenever available) facilitate accurate gross tumor delineation. The commonly used strategy is centered on contouring the GTV with a 0 mm margin expansion to create the CTV. An elective dose CTV to include a concentric expansion of the GTV or to encompass a limited elective nodal volume is at the discretion of the treating radiation oncologist. The PTV is a uniform expansion of 3 to 5 mm from the GTV/CTV based on institutional practice [12]. 

### 3.2. SBRT Dose and Fractionation 

Dose prescription varied across institutions and ranged from 12 to 22 Gy single fraction, 24 to 25 Gy/2 fractions, 24 to 27 Gy/3 fractions, 24 to 30 Gy/4 fractions, and 30 to 50 Gy/5 fractions, with BED_10_ range from 26.4 to 100 Gy_10_. The primary factors influencing the selection of fractionation schedules often encompass tumor size, site, close proximity to critical structures, previous radiation doses administered, and an indication for SBRT [24]. Treatment was often delivered either every other day or twice weekly, 2 days apart.

### 3.3. Target Objectives and OAR Constraints 

Plan normalization should provide coverage of ≥95% of the PTV. Planning optimization uses conformity indices, D95%, D99%, near-minimum dose (D98%), and near-maximum dose (D2%) [24]. Critical OARs are the spinal cord, brain, brainstem, optic chiasm, optic nerves, and eyes. Table 1 summarizes the dose constraints for various SBRT fractionation regimens. Patients are to be planned and treated using IMRT or VMAT planning (ideally with a ≤5 mm leaf width of the multi-leaf collimator). Maximum point dose up to 115% of the prescription dose is acceptable within the PTV and the prescription dose outside of the PTV should be avoided. The aim is to achieve a conformality index (CI) < 1.1. A daily cone beam computed tomography (CBCT) should be performed with pre- and post-shifts, with a physician present at day 1 of SBRT treatment.

## 4. Definitive SBRT for Primary HNC 

In general, SBRT is used in the palliative setting for HNC patients who are unable to attend standard long courses of RT (e.g., social and logistic challenges), and when omission or significant reduction of the elective target volume is clinically acceptable. This includes the following clinical scenarios: (1) SBRT for elderly/medically unfit patients aiming to maximize locoregional control (LRC) and decrease the disease burden for HNC, (2) SBRT for early glottis cancer, or (3) SBRT boost to gross tumor volume (GTV) after definitive external beam radiation therapy (EBRT) as an alternative option to brachytherapy boost.

### 4.1. Definitive SBRT in Elderly or Medically Unfit HNC Patients 

The ultimate goal of SBRT in elderly or medically unfit HNC patients is to achieve an acceptable balance between LRC, cancer-associated disease burden, and RT-related toxicity [14,16,20,21,22,23,32,33]. SBRT demonstrated acceptable local control (LC) rates with minimal side effects compared to conventional fractionation RT with a standard comprehensive target volume [15]. The literature included single-institution studies varied in number of included patient (3–106 patients), primary tumor sites and SBRT doses and fractionation schedules (15–22 Gy in single fraction to 30–50 Gy in five or six fractions (BED_10_ range between 32.17 and 91.65 Gy_10_)) [24]. The one-year LC and overall survival (OS) rates ranged from 69% to 87% and 60% to 85%, respectively [12,14,16,20,21,22,23,32,34]. Acute or late grade 3 toxicities included osteoradionecrosis, pain, dermatitis, ulceration, and cataracts [12,20,22,33,34] (See Table 2). 

A meta-analysis evaluated SBRT for de novo HNC in elderly patients who could not undergo aggressive CRT or altered fractionation RT (median age: 76 years). SBRT dose ranged from 25 to 59.5 Gy in 3 to 17 fractions, with a median BED_10_ ranged from 42.63 to 82.72 Gy_10_ and an equivalent dose in 2 Gy fractions (α/β = 10) between 35.53 and 68.93 Gy. The 3-year LC rate was acceptable (73.5%), and the 3-year OS was approximately 50%, indicating that the focus might have been on optimizing the LC rather than OS due to comorbidities and old age of those patients. The late grade 5 toxicity rate was 0.1% [35]. 

#### Summary and Recommendation 

There is limited evidence supporting the use of definitive SBRT for elderly or medically unfit HNC patients who cannot tolerate a standard long course of RT. A wide SBRT dose range was used (15 to 22 Gy in 1 fraction to 30 to 50 Gy in 5–6 fractions). Further studies are warranted to establish the optimal SBRT dose, fractionation, and criteria for selecting patients with primary HNC for definitive SBRT. 

### 4.2. Definitive SBRT for Early-Stage Glottis Cancer 

The use of SBRT is considered an attractive treatment option for early-stage glottis cancer given the shorter overall treatment time associated with SBRT that could potentially improve the LC. In addition, there is no need to treat the un-involved contralateral vocal cord or elective nodal target volume which allows a higher dose per fraction without possibly significant late morbidity [36,37,38,39].

A phase I trial from the University of Texas Southwestern Medical Center investigated 3 dose levels (50 Gy/15 fractions, 45 Gy/10 fractions, and 42.5 Gy in 5 fractions) for 29 patients with early (Tis-T2) glottis cancer (median follow up: 39.2 months). Two patients had dose-limiting toxicity: one with cT2 cancer received 45 Gy in 10 fractions, who developed grade 4 laryngeal edema and grade 3 dysphagia at 5 months post-RT, and another patient with cT2 disease treated with 42.5 Gy in 5 fractions developed grade 3 laryngeal necrosis and grade 3 dysphagia at 7 months post-RT [40]. The voice handicap index improved in all groups. A total of 5 patients developed recurrence (no recurrence was observed in the 42.5 Gy group). Although there were 2 dose-limiting toxicities; these results were the foundation of an ongoing phase II trial (NCT03548285) investigating two SBRT schedules based on risk groups: low-risk (PTV < 10 cc and no smoking within 1 month from registration: SBRT with 42.5 Gy/5 fractions) and moderate-risk (PTV >10 cc, or smoking history within 1 month from the registration [≤1 pack/day]: RT with 58.08/16 fractions) [41].

Another phase I trial for early glottis cancers evaluated 59.5 Gy/17 fractions and 55 Gy/11 fractions. The initial report showed satisfactory toxicity levels and favorable voice/quality of life (QoL) outcomes [42]. However, Kang et al.’s update led to the trial closure due to toxicity in the 55 Gy group (arytenoids necrosis at 5 months post-SBRT, and vocal cord ulcer at 15 months post-SBRT), following predefined stopping rules [43]. The authors concluded that SBRT is not feasible for early glottis cancer [43]. 

#### Summary and Recommendation

Two phase I trials evaluated SBRT for early glottis cancer and showed the development of pre-defined dose limiting toxicities. An ongoing phase II trial is evaluating the potential use of risk-adaptive SBRT dose selection in the setting of SBRT for early glottis cancer. SBRT twice a week for T1/T2 lesions is an interesting option, acknowledging the risk of severe late toxicity, including chondronecrosis, which may be dependent on pre-existing infiltration of the laryngeal framework. 

### 4.3. Definitive SBRT as Boost after EBRT (Alternative to Brachytherapy Boost)

In 2008, Hara et al. updated results from Tate et al. (1999) [44] and Lee et al. (2003) [45] on SBRT boost for 82 patients (47 had stage IV nasopharynx cancer). SBRT boost of 7–15 Gy was given for 2–6 weeks after EBRT. At 5 years, local failure, regional failure, DM rates, and OS were 2%, 17%, 32% and 69%, respectively. The late toxicities included radiation-induced retinopathy (n = 3), carotid aneurysm (n = 1), and temporal lobe necrosis (n = 10) [46]. Chen et al. also reported outcomes and toxicity of SBRT boost (12–15 Gy in 4–5 fractions) to nasopharynx cancer (n = 64). The 3-year LC rate was 93.1%. Three patients had fatal nasal bleeding at 6–7 months after SBRT boost [47]. 

Uno et al. investigated the feasibility of SBRT boost (9–16 Gy in 1–3 fractions) for various HNC sites in 10 patients [48]; 60% had a complete response (CR), 40% had a partial response (PR), with no grade ≥ 3 toxicities attributable to SBRT. In a Japanese series of 25 HNC patients treated with SBRT boost (12–35 Gy in 1–5 fractions), 18 patients had CR, 6 patients had PR, and one patient with disease progression (DP), which resulted in a 96% (24/25) overall response rate (ORR). The 2-year LC and OS rates were 89% and 70%, respectively. The small SBRT planning target volume (PTV) boost (≤20 cm^3^) and the good initial response to RT predicted favorable outcomes in terms of LC and OS [49]. 

Lee et al. evaluated the long-term outcomes and toxicity of SBRT boost (10–25 Gy in 2–5 fractions) in 26 HNC patients. The major response rate was 100% (21 CR). A total of 9 patients experienced grade ≥3 toxicities, of whom, 5 patients with late grade 3 (including pontine necrosis, temporal lobe necrosis (n= 2), radiation retinopathy, neovascular glaucoma, and optic neuropathy), 4 patients with late grade 4 toxicity (including soft tissue necrosis in the left base of the skull bone, mucosal ulcer and necrosis, soft tissue necrosis in the left nasopharyngeal wall, and an unhealed mucosal ulcer with bleeding), and 1 patient with grade 5 pontine necrosis. SBRT boost volume (median 47.7 cc) predicted late complications [50].

Almamgani et al. prospectively evaluated SBRT boost (16.5 Gy in 3 fractions) for 51 patients with stage I-IVb oropharyngeal carcinoma (OPC), not suitable for a standard brachytherapy boost [51]. The reported 2 year LC and OS rates were 86% and 82%, respectively, with acceptable toxicity including feeding tube dependency (n = 1) and grade 3 xerostomia (n = 2). Following this, they implemented the same treatment for cases of T1–2 and selected small T3, N0-N2 OPC. They documented the incidence of treatment failure, treatment results, and long term treatment related toxicity in a group of 195 patients who received treatment between 2009 and 2016 [52,53]. The reported 5 year LC and regional control (RC) rates were 90% and 93%, respectively. By location of the center of the recurrent disease, 76% of the failures were within the treated volume and 24% were outside the treated volume. This is notably higher than what has been reported in the existing literature, and it is attributed to the highly delivered dose escalation [52]. With a median follow-up of 4.3 years, the reported 5 year disease specific survival (DSS) and OS rates were 85% and 67%, respectively. Notably, severe (grade ≥3) toxicity was reported in 28% of patients, with the most common adverse effects being dysphagia, weight loss, mucosal ulceration, soft tissue, and osteoradionecrosis [53]. 

In a phase I trial of dose-escalated SBRT boost to residual gross tumor of 8 or 10 Gy in a single fraction, or 10 Gy in 2 fractions, after 60–66 Gy/30–33 fractions with concurrent cisplatin for unfavorable intermediate- or high-risk OPC, the LC rate was 85.3% at 4.3 years. Four patients with tumor necrosis had grade 3 dysphagia, and three patients had grade 4 pharyngeal hemorrhage requiring surgical intervention [54]. The outcome, patterns of failure, and toxicity profile of various SBRT boost studies are described in Table 3. 

#### Summary and Recommendation 

Despite an acceptable oncologic outcome of SBRT boost after EBRT for HNC, severe treatment-related toxicities have been reported. As such, the use of SBRT boost for HNC as an alternative to brachytherapy boost is recommended only in the investigational setting.

## 5. Neoadjuvant SBRT (with Immunotherapy) for HNC

Immunotherapeutic approaches are effective in recurrent/metastatic HNC [57] and enhance treatment when combined with other modalities [58]. SBRT can overcome immunotherapy resistance and sensitize cancer cells [59]. Neoadjuvant immunoradiation could potentially improve the oncologic and functional outcomes through shortening the overall treatment time, limiting radiation target volumes, and facilitating less extensive surgery through downsizing the tumor [60]. 

A phase Ib/II trial included 19 patients (phase Ib: 6; phase II: 13) with untreated locally advanced HPV-related OPC. Patients received neoadjuvant durvalumab ± tremelimumab for 2 doses (durvalumab only [n = 3]; durvalumab + tremelimumab [n = 16]), with concurrent SBRT of 25 Gy in 5 fractions to gross disease only, followed by transoral robotic surgery with adjuvant durvalumab for up to 4 cycles. The median follow-up was 12.7 months. No safety signals were reported. A total of 18 out of 19 patients (95%) achieved a clinical/pathological downsizing, of whom 9 (47%) had a pathologic complete response (pCR). In total, 5 patients (26%) developed locoregional failure (LRR), with a median time to recurrence of 3 months. Failing to achieve pCR was significantly associated with LRR (*p* = 0.03). Caution against omitting elective volume irradiation is warranted even in a favorable prognosis HPV-related OPC in the neoadjuvant setting with SBRT and immunotherapy [61]. 

In a phase Ib trial, locally advanced p16-positive and p16-negative HNSCC patients were treated with neoadjuvant SBRT over 1 week with nivolumab (240 mg intravenous q2 week’s ×3 cycles) before surgery. Cohort-I included 5 patients who received 40 Gy in 5 fractions; cohort-II included 5 patients who received 24 Gy in 3 fractions. After assessment of the toxicity, 2 expansion cohorts were added: cohort-III which included 6 patients who received SBRT alone (24 Gy in 3 fractions) for stages I-III HPV-related HNSCC and cohort-IV included 5 patients who received nivolumab + SBRT (24 Gy in 3 fractions) for stages III-IVA p16-negative HNSCC. Surgery was scheduled for 5 weeks post-SBRT, followed by adjuvant nivolumab 480 mg intravenous q4 weeks for 3 doses starting 4 weeks after surgery in all cohorts. All 21 patients completed neoadjuvant treatment without dose-limiting toxicity. In the entire study group, the major pathological response (mPR) and pCR rates were 86% and 67%, respectively. Among the 10 HPV-related HNSCC patients who underwent treatment with nivolumab and SBRT, the pCR rate was 90% (cohort-I = 5/5; cohort-II = 4/5) and the mPR rate was 100%. In HPV-related HNC patients treated with neoadjuvant SBRT alone (cohort-III), the pCR rate was 50% (n = 3). In HPV-negative patients (cohort-IV), the pCR and mPR rates were 20% (n = 1) and 60% (n = 3), respectively [60].

A phase I/Ib trial was conducted to evaluate the safety of administering both SBRT and a single dose of durvalumab as neoadjuvant treatment for 21 patients with HPV-unrelated locally advanced HNSCC [62]. Patients received neoadjuvant durvalumab (1500 mg) and SBRT approximately 3–6 weeks before surgery. The starting SBRT dose level was 6 Gy for 2 fractions (12 Gy total) every other day to gross disease. If the dose was tolerated, the dose was increased to 6 Gy for 3 fractions (18 Gy total) for the next 3 patients then 6 Gy for 4 fractions (24 Gy total). Adjuvant therapy was used based on a standard of care indications for the first enrolled 8 patients, and all patients received adjuvant durvalumab to be initiated approximately 6–12 weeks post-surgery. It was given as 1500 mg intravenously once every 4 weeks for a maximum of six doses, or until disease progression, unacceptable toxicity, or withdrawal from the study. The protocol was updated after the 8th enrolled patient to omit adjuvant RT for patients with pCR or mPR, but all patients still received adjuvant durvalumab. The safety endpoint was met. With a median follow-up of 16 months, OS was 80.1%, LRC and PFS were 75.8%, and mPR was 75%. For patients treated with 24 Gy in 4 fractions, the mPR rate was 89%. Radiation dose and time from SBRT to surgery correlated with mPR. One patient, treated below the maximum tolerated dose, recurred out of the SBRT volume, despite having received adjuvant RT and durvalumab. Two other patients failed in the SBRT volume, of whom one refused adjuvant RT but received adjuvant durvalumab [62].

Shen et al. retrospectively studied 30 locally advanced oral cavity SCC patients treated with neoadjuvant nivolumab plus SBRT (median dose: 24 Gy, range, 14–48 Gy) with 56.6% of patients receiving adjuvant RT +/− chemotherapy. Treatment was well-tolerated with no serious adverse events. R0 resection was achieved in 90% of patients, with 16.7% of patients experiencing procedure-associated complications. Response rates were CR 10%, PR 46.7%, and SD 43.3%. The mPR and pCR rates were 60.0% and 33.3%, respectively. Median follow-up was 13.5 months. The 2-year disease-free survival (DFS) and OS were 70.4% and 76.4%, respectively, for 26 patients with surgical resection. Patients with mPR and CR showed significantly better DFS and OS (*p* < 0.05) [63].

### Summary and Recommendation 

Neoadjuvant SBRT with immunotherapy is a safe treatment for locoregionally advanced HNSCC, potentially resulting in relatively high rates of mPR with subsequent favorable outcomes. The commonly used SBRT regimen in the neoadjuvant setting is 24 Gy/3 fractions and 25–40 Gy in 5 fractions. Omitting elective nodal irradiation during neoadjuvant SBRT has a higher risk of regional nodal recurrence even in favorable HPV-related OPC despite the use of immunotherapy. Futures studies are warranted to further confirm the efficacy of this strategy [60,61,62,63].

## 6. Salvage SBRT for Recurrent Unresectable or Second Primary HNC

Salvage SBRT for unresectable recurrent and second primary HNC in a previously irradiated volume is challenging. While studies consistently demonstrate improved LC with re-irradiation, the accumulation of high cumulative doses may result in severe side effects, such as the potentially fatal carotid blowout syndrome. Hence, it is crucial to carefully select patients and appropriate RT techniques [17,20,64,65,66,67,68,69,70,71,72].

Heron et al. conducted a phase I dose-escalation trial with salvage SBRT for recurrent HNC. A total of 25 participants received escalating SBRT doses, starting at 5 Gy per fraction that was escalated to 8.8 Gy per fraction for 5 fractions delivered over 2 weeks. The maximum tolerated dose was 44 Gy in 5 fractions, with no associated grade ≥ 3 acute toxicities, and an ORR of 17%, a median duration of response of 4 months, and a median OS of 6 months [73]. An updated report included 85 patients and showed that SBRT doses ≥35 Gy resulted in improved LC (71% vs. 59%, *p* = 0.01). The 1-year and 2-year LC and OS rates were 51.2% and 30.7%, and 48.5% and 16.1%, respectively [72]. 

A retrospective-matched case-control study investigated concurrent cetuximab with SBRT (n = 35) vs. SBRT alone (n = 35) for unresectable recurrent HNSCC. Both study arms received a median SBRT dose of 40 Gy (range, 20–44 Gy). Concurrent cetuximab showed improved OS (median 24.5 vs. 14.8 months, *p* = 0.03) [74]. In 2014, an updated retrospective review included 132 patients who were treated with salvage SBRT for recurrent HNC, with a median dose of 44 Gy in 5 fractions (range, 35–50 Gy), and a median follow-up of 6 months [17]. The 1-year OS and LRC rates were 38% and 48%, respectively. Overall, the toxicity rates were acceptable; 16 patients (12%) and 6 patients (7%) experienced grade ≥ 3 acute and late toxicity, respectively (with the majority of toxicity related to mucosal and skin reactions) [17]. Treatment duration < 14 days improved recurrence-free survival but increased late toxicity (*p* = 0.03). This study found that tumor volume > 25 cc predicted inferior survival, poor tumor control, and more acute toxicity (*p* = 0.02) but no difference in late toxicity [17]. 

Comet et al. conducted a phase I trial investigating the use of salvage SBRT with or without cetuximab for patients who developed local recurrence or new primary HNC [69]. In this trial, a total of 40 patients with 43 lesions received 36 Gy in 6 fractions, SBRT treatment, with 15 of them (37.5%) undergoing concurrent cetuximab, and 1 patient receiving concurrent cisplatin [69]. Half of the patients had HNSCC. The 1-year OS rate was 58%. Among the 34 patients assessed for treatment response, 15 (44%) had CR, 12 (35%) had PR, and 7 (21%) had SD. Notably, among the 14 patients who received concurrent cetuximab, 75% achieved an overall objective response [69]. Subsequently, Lartigau et al. conducted a phase II multi-institutional trial to evaluate re-irradiation using salvage SBRT, combined with concurrent cetuximab, in 56 patients diagnosed with recurrent or new primary HNSCC. These patients received 36 Gy delivered in 6 fractions over 11 to 12 days [70]. The 1-year OS was 47.5% [70]. Of the 49 evaluable study participants, the ORR was 69%; CR was seen in 24 (49%), PR in 10 (20%), and SD in 11 (23%). Notably, 18 of the study patients (32%) encountered severe toxicities rated at grade ≥ 3 and 1 treatment related in death because of an arterial rupture [70]. These findings align with those reported in Heron et al.’s study [74]. Lartigau et al. [70] attributed the low rate of carotid blowout events to the cautious identification of patients with recurrent or new primary HNC without tumor encasement involving lower than third of the carotid artery. 

Cengiz et al. retrospectively analyzed 46 patients with locally recurrent HNC (65% had HNSCC) treated with re-irradiation using SBRT (median dose: 30 Gy, range: 18–35 Gy, 1 to 5 fractions) [68]. The 1-year OS rate was 46% [68]. Out of the 37 study patients assessed for response, 10 (27%) achieved CR, 11 (30%) demonstrated PR, and 10 (27%) had SD. Despite the comparable survival outcome with other studies [69,70], the study reported a higher incidence of late-grade ≥ 4 toxicity, with 8 patients (17%) experiencing late carotid blowout, of whom 7 died from a carotid hemorrhage [68]. It has been suggested that the relatively elevated rate of late toxicity in the study might be attributed to the daily SBRT fractionation schedule, rather than an every-other-day SBRT fractionation schedule employed in other studies [17]. 

Unger et al. reviewed 65 patients treated with SBRT for recurrent HNC. The study included 27 patients (42%) with metastatic disease or untreated local disease, 11 (17%) with non-squamous histologies, 19 (29%) treated with surgery prior to re-irradiation, and 21 (32%) treated with CRT. The SBRT dose ranged from 21 to 35 Gy in 2 to 5 fractions [64]. The group reported an ORR of 80%; CR rate of 54%, and PR rate of 27%. The median OS was 12 months and the 2-year OS rate for patients without metastatic cancer was 41%. Seven patients (11%) experienced late toxicities related to SBRT, and 1 patient died due to treatment [64]. Roh et al.’s reviewed 36 patients with 44 lesions, all of whom had local recurrence and were treated with SBRT with dose ranging 18 to 40 Gy (median, 30 Gy) in 3 to 5 fractions [71]. More than half of the lesions were SCC. Median OS was 16 months, with CR rate of 43%, PR rate of 37%, and SD in 9%. Grade 3 acute complications affected 36% of participants, and late complications affected 8%. The study reported a notably high incidence of late grade ≥4 toxicities, which some attributed to daily radiation rather than every-other-day delivery [17,71].

Vargo et al. studied 414 patients with unresectable recurrent or second primary HNC treated with intensity-modulated radiation therapy (IMRT, n = 217 patients) or SBRT (197 patients). The OS was similar for IMRT and SBRT with dose ≥35 Gy for small tumor volumes (25 cc); however, dose < 35 Gy resulted in significantly worse 2-year OS of 14% [15]. Another study with 45 patients showed higher 1-year OS of 68% with ≥40 Gy in 5 fractions, compared with 24% with lower doses [75].

### Summary and Recommendation

Salvage SBRT for recurrent (or 2nd primary) HNC in previously irradiated volume showed acceptable survival (Table 4) [17,64,65,71]. Rate of carotid blowout is relatively low with appropriate patient selection, target volume definition, and every-other-day treatment delivery. However, differences in patient selection criteria, tumor histology, and salvage SBRT doses make direct comparisons challenging. Therefore, a large, multi-institutional trial for re-irradiation using SBRT is warranted.

## 7. Adjuvant SBRT for Recurrent HNC

An ongoing multi-center phase II trial (STEREO POSTOP, NCT03401840) evaluates post-operative SBRT (36 Gy in 6 fractions over 11–13 day) for pT1-2 N0-1 oral cavity SCC and OPC with compromised resection margins (with no pathologic extranodal extension) [76]. The study hypothesizes that postoperative SBRT’s safety and efficacy will be similar to a conventional RT schedule [77,78].

Vargo et al. [79] conducted a retrospective study on 28 patients who had high-risk features (involved resection margin(s) or pathologic extranodal extension) following salvage surgery with gross total resection (i.e., R0/R1) followed by adjuvant SBRT with (7/28 patients) or without (11/28) cetuximab. The SBRT dose was 40 to 44 Gy in 5 fractions over 1–2 weeks. All patients had previously received RT (median dose of initial RT was 70 Gy; range, 54–99 Gy), with a median time to re-irradiation (from original RT) of 25 months (range, 6–156 months). Median follow-up was 14 months (range, 2–69 months). The 1-year LRC, distant control, DFS, and OS rates were 51%, 90%, 49%, and 64%, respectively. The rates of acute and late severe (grade ≥ 3) toxicity were 0% and 8%, respectively [79]. At six months follow-up, 56% of patients reported improved or stable overall QoL scores [79].

## 8. Conclusions

Head and neck SBRT represents a significant advancement in the field of radiation therapy, offering a promising treatment option for highly selected patients with HNCs who are not suitable for standard treatment options. Multidisciplinary case discussion, close monitoring, and follow-up are crucial to assess treatment response and manage any potential treatment related side effects.

## 9. Future Directions

Recent advances in immunotherapeutic agents showed promising outcomes in the treatment of HNC. The combined application of these drugs alongside SBRT is currently under active research. For example, the RTOG 3507 phase II clinical trial is exploring the use of re-irradiation with SBRT plus concurrent pembrolizumab for patients with recurrent HNSCC in a previously irradiated volume [80]. Furthermore, recent advances in RT technology such as magnetic resonance-guided radiation therapy (MRgRT) for HNCs allows precise treatment, facilitates tighter PTV margin/smaller irradiated volumes, evaluates tumor response with functional imaging, i.e., DWI, with possibly response-adaptive RT. However, further research is required for the evaluation of predictive MR imaging biomarkers, and the use of SBRT with MRgRT for patients with HNC who cannot tolerate long course RT [81]. Moreover, the impact of SBRT for HNC in the palliative setting aiming to improve HNC outcomes in patients who are unable to tolerate curative-intent RT is going to be investigated by the CCTG HN13 phase III randomized controlled trial (SBRT vs standard palliative RT).

## Figures and Tables

**Table 1 cancers-15-05010-t001:** Organs-at-risk constraints among different head and neck SBRT regimen.

OAR Constraint	Constraint for 1 fx		Constraint for 2 fx		Constraint for 3 fx		Constraint for 4 fx		Constraint for 5 fx		Endpoint ≥ Grade 3	
	Primary Disease	Re-RT	Primary Disease	Re-RT	Primary Disease	Re-RT	Primary Disease	Re-RT	Primary Disease	Re-RT	Primary Disease	Re-RT
Spinal cord and medulla_ PRV	Dmax 14 Gy (D0.035cc), V10 (<0.35cc) [26,27,28,29]	Dmax 9 Gy [26,30]	Dmax 17–19.3 Gy (D0.035cc), V13 (<0.35cc) [30]	Dmax 12.2 Gy [26,30]	Dmax 20.3–22.5 Gy (D0.035cc), V15.9 (<0.35 cc) [26,27,30]	Dmax 14.5 Gy [26]	Dmax 23–25.6 Gy (D0.035cc), V19.2 (<0.35 cc) [26,29]	Dmax 16.2 Gy [26,30]	Dmax 25.3–30 Gy (D0.035cc), V22 (<0.35 cc) [26,27,30]	Dmax 18 Gy [26,30]	Myelitis [29] Sahgal et al. [26]: Radiation myelopathy (1–5% risk for 1–5 fractions)	Myelitis [30]
Optic pathway	Dmax 10 Gy, V8 (<0.2 cc) [29]	Dmax 8 Gy [24]	Dmax 17.3 Gy, V11.7 (<0.2 cc) [29]	-	Dmax 17.4 Gy, V15.3 (<0.2 cc) [29]	Dmax Gy, V15 < 0.2 cc (Optic nerves) [24]	Dmax 21.2 Gy, V19.2 (<0.2 cc) [29]	-	Dmax 25 Gy, V23 (<0.2 cc) [29]	Dmax 10 Gy [24]	Neuritis [29]	-
Cochlea	Dmax 10 Gy [29], Dmax 4–12 Gy [24]	Dmax 12 Gy [24]	Dmax 13.7 Gy [29]	-	Dmax 17.4 Gy [29], Dmax 20 Gy [24]	Dmax 24 Gy [24]	Dmax 21.2 Gy [29]	-	Dmax 22 Gy [29], Dmax 25–30 Gy [24]	Dmax 20–27.5 Gy [24]	Hearing loss [29]	-
Brain stem (not medulla)	Dmax 15 Gy, V10 (<0.5 cc) [29]	Dmax 10–15 Gy, V10 < 1 cc [24]	Dmax 17.3, V13 Gy (<0.5 cc) [29]	-	Dmax 23.1 Gy, V15.9 (<0.5 cc) [29]	Dmax 23 Gy, V18 < 1 cc [24]	Dmax 27.2 Gy, V20.8 (<0.5 cc) [29]	-	Dmax 31 Gy, V23 (<0.5 cc) [29]	Dmax 9–15 Gy [24]	Cranial neuropathy [29]	-
Esophagus	Dmax 24 Gy, V20 (< 5 cc) [29], Dmax 19 Gy [24]	Dmax 10 Gy [24]	Dmax 28.3 Gy, V24.3 (<5 cc) [29]	-	Dmax 32.4 Gy, V27.9 (<5 cc) [29]	-	Dmax 35.6 Gy, V30.4 (<30.4 cc) [29]	-	Dmax 38 Gy, V32.5 (5 cc) [29], Dmax 27–35 Gy [24]	Dmax 20–25 Gy [24]	Esophagitis [29]	-
Brachial plexus	Dmax 16.4 Gy, V 13.6 (<3 cc) [29]	Dmax 10–16 Gy, V14.4 <3 cc [24]	Dmax 20.8 Gy, V17.8 (<3 cc) [29]	-	Dmax 26 Gy, V22 (<3 cc) [29]	Dmax 23 Gy, V22.5 <3 cc [24]	Dmax 29.6 Gy, V24.8 (24.8(3 cc) [29]	-	Dmax 32.5 Gy, V27 (3 cc) [29]	Dmax 20–32 Gy V30 < 3 cc [24]	Neuropathy [29]	-
Trachea	Dmax 30 Gy, V27.5 (<4 cc) [29]	-	Dmax 38 Gy, V34.5 (<4 cc) [29]	-	Dmax 43 Gy, V39<(5 cc) [29]	-	Dmax 47 Gy, V42.4(5 cc) [29]	-	Dmax 50 Gy, V45 (<5 cc) [29]	-	Stenosis [29]	-
Skin	Dmax 27.5 Gy, V25.5 (10 cc) [29]	-	Dmax 30.3 Gy. V28.3 (10 cc) [29]	-	Dmax 33 Gy, V31 (10 cc) [29]	-	Dmax 54 Gy, V33.6 (10 cc) [29]	-	Dmax 38.5 Gy, V36.5 (10 cc) [29]	Dmax 20 Gy [24]	Ulceration [29]	-
Brain	V12 Gy (10–15 cc) [31], Dmax 15–20 Gy V10 < 1 cc [24]	Dmax 10 Gy [24]	-	-	20 Gy (D20cc) [31], Dmax23 Gy V18 < 1 cc [24]	-	-	-	24 Gy (D20cc) [31], Dmax 10–25 Gy [24]	Dmax 20–23 Gy [24]	Milano et al. [31]: Symptomatic radiation necrosis (one fraction), oedema/necrosis (three and five fractions)	-
Carotid artery	-	Dmax 10 Gy [24]	-	-	-	-	-	-	Dmax 25–47 Gy [24]	Dmax 15–34 Gy < 50% gets PTV dose [24]	-	-
Parotid	-	-	-	-	-	-	-	-	-	Dmax 20–25 Gy [24]	-	-
Lens	-	-	-	-	-	-	-	-	-	Dmax 6 Gy [24]	-	-
Larynx	-	-	-	-	-	-	-	-	Dmax 20 Gy [24]	Dmax 20 Gy [24]	-	-

Abbreviations: Dmax: Maximal dose, Fx: Fraction, OAR: Organ-at-risk, Re-RT: Re-irradiation.

**Table 2 cancers-15-05010-t002:** Summary of retrospective SBRT studies for primary head and neck cancer.

Author (Year)/Design/Subsite	n	Median Age (Range), yr	Median Target Volume	Elective Nodal Irradiation	RT Dose (Gy)/Fraction	EQD_2_ (Gy) (α/β = 10)	BED_10_ (Gy) (α/β = 10)	BED_3_ (Gy) (α/β = 3)	Median Follow Up (Months)	LC (%)	OS (%)	Toxicity
Voruganti et al. (2021)/retrospective/skin [33]	106	86 (56–102)	(GTV) = 31 cm^3^ (range: 17–56 cm^3^)	Yes	32–50/4–6	48–76.38	57.6–91.65	117.3–188.83	8	1 yr 78%	1 yr 53%	Acute: Grade 3: 31 dermatitis Late grade ≥ 3: 7 fibrosis, 1 ORN and 1 grade 4 skin ulceration
Al-Assaf et al. (2020)/retrospective/mixed [12]	48	81 (25- 102)	Median GTV volume = 33.2 cc (range, 1.9–368.6 cc)	Yes	35–50/4–6	54.69–76.38	65.63–91.65	137–189	10.5	85.5%	-	Acute: Grade 4:1 (Mucosal ulceration) Late: Grade 4:1 (ORN and skin ulceration)
Gogineni et al. (2020)/retrospective/mixed [34]	66	80 (47–99)	Median PTV volume = 82 cc	Yes	35–40/5	49.58–60	59.5–72	116.67–146.67	15 (3–88)	1 yr 73%	1 yr 64%	Acute: Grade 3:2 Late: Grade ≥ 3:0
Khan et al. (2015)/retrospective/mixed [14]	17	87 (25–103)	Median Maximum Diameter = 3.7 cm (1–10 cm)	Yes	35–48/5–6	49.58–72	59.5–86.4	116.67–176	8	1 yr 87%	1 yr 60%	Grade 3:0
Amini et al. (2014)/retrospective/mixed [16]	3	82 (72–88)	Tumor volume cc = 15–36.7 cc	Yes	25–36/5	31.25–51.6	37.5–61.92	66.67–122.4	8	100 (crude rate)	33	Grade 3 = 0
Vargo et al. (2014)/retrospective/mixed [17]	12	88 (79–98)	Median = 42.1 cc (15.1–247.9 cc)	No	20–44/1–5	50–68.93	60–82.72	155.33–173.07	6 (0.5–29	1 yr 69%	1 yr 64%	Acute: Grade 3:1 Late: Grade 3:1
Kawaguchi et al. (2012)/retrospective/mixed [22]	14	73 (64–93)	-	No	35–42/3–5	63.18–64.4	75.81–77.28	171–77.28	36 (14–40)	Mean 71.4	Mean 78.6	Late: Grade 3:1 (ORN) (after 2nd SRS)
Karam et al./retrospective/ parotid [32]	13	80 (34–99)	PTV = 13.3–195.3 cc	Yes	25–40/5–7	31.25–52.37	37.5–62.84	66.67–116.13	14 (0–59)	2 yr LRC 84%	2 yr 46%	Acute: G5: 1 Sepsis secondary to aspiration pneumonia
Kodani et al. (2011)/retrospective/mixed [21]	13	66 (17–88)	Median GTV volume = 22 cc (0.7–78 cc)	No	19.5–42/3–8	26.81–53.38	32.17–64.05	61.75–115.5	16 (3–51)	CR:38% PR:46%	85%	Grade 3:0
Siddiqui et al. (2009)/retrospective/mixed [20]	10	73.5 (37–89)	Median GTV 15.5 cc (1.7–155 cc)	No	30–48/5–6	40–72	48–86.4	90–176	32 (7–53.4)	1 yr 83.3%	1 yr 70%	Acute: Grade 3:1 (Pain) Late: Grade 3:1 (Cataract)

Abbreviations: RT: Radiotherapy, EQD2: Equivalent dose at 2 Gy/fraction, BED_10_: Biologically effective dose (α/β = 10); BED_3_: Biologically effective dose (α/β = 3) LC: local control, OS: Overall survival, GTV: Gross tumor volume, PTV: Planning target volume, CR: Complete response, PR: Partial response, ORN: Osteoradionecrosis.

**Table 3 cancers-15-05010-t003:** Summary of SBRT boost studies in head and neck cancer.

Author (Year)/Subsite/Design	Sample Size (n)	Median Follow Up (Months)	EBRT Dose/Fraction	Boost Dose (Gy)/Fraction	GTV (cc) or Boost Volume (Range)	EQD2 (Gy) (α/β = 10) (Total)	BED10 (Gy) (α/β = 10) (Total)	Margins for Stereotactic Boost (PTV)	LC (%)	OS (%)	Initial Site of Failure (N)	Toxicity (N)
Tate et al. (1999)/retrospective/nasopahrynx [44]	23	21 (2–64)	64.8 Gy–70 Gy (Median 66 Gy/ 33 frs)	7–15 Gy /1#frs Median 12 Gy	Not reported	Median 88	Median 105.6	Not reported	100%	Not reported	Local: 0 Regional: 2 Distant: 7	As expected for EBRT
Le et al. (2003)/retrospective/nasopahrynx [45]	45	31	66 Gy/33 frs	7–15 Gy/1 frs	Not reported	88	105.6	Not reported	3 yr LC: 100%	3 yr OS: 75%	Local: 0 Regional: 3 Distant: 14	CN weakness: 4 Retinopathy: 1 Asymptomatic TLN: 3
Chen HH et al. (2006) retrospective/nasopahrynx [47]	64	31 (22–54)	64.8 Gy–68.4 Gy/36–38 frs	12–15 Gy/4–5 frs	Mean GTV 62.6 (21.1– 145.3)	76.72–83.51	92.06–100.2	CTV + 2–3 mm	3 yr LC: 93.1%	3 yr OS: 84.9%	Local: 4 Regional: 7 Distant: 7	Late Grade 4: None Note: 3 fatal nasal bleeding could be not related to SBRT boost
Hara et al. (2008)/retrospective/nasopahrynx [46]	82	40.7 (6.5–144.2)	66 Gy/33 frs	7–15 Gy/1 frs	Median GTV 34.2 (6.4– 102.2)	88	105.6	Not reported	5 yr LC: 98%	5 yr OS: 69%	Local: 1 Regional: 5 Distant: 27	Retinopathy: 3 Asymptomatic TLN: 8 Symptomatic: 2
Uno T et al. (2010)/retrospective/mixed [48]	10	16 (6–24)	40 Gy–60 Gy/20–30 frs	9–16 Gy/1–3 frs	Not reported	54.22–80.44	65.1–96.53	CTV + 0–5 mm	CR: 60% PR: 40%	Not reported	Local: 3 Distant: 1	≥Grade 3: None
Lee DS et al. (2012) retrospective/mixed [50]	26	56 (27.6–80.2)	39.6 Gy–70.2 Gy (Median 50.4 Gy/28 frs)	10–25 Gy/2–5 frs Median 21 Gy/5 frs	NPC median GTV 45.3 (21.3– 69.4) Non-NPC Median GTV 19.4 (6.9–66.8)	Median 74.41	Median 89.29	GTV + 1- mm	1 yr LRRFR: 91.4% 2 yr LRRFR: 86.3%	2 yr OS: 61.5% 5 yr OS: 46.2%	Local: 2 Regional: 1 Distant: 5	≥Grade 3: 9
Al-Mamgani et al. (2012)/retrospective/oropharynx [51]	51	18 (6–65)	46 Gy/23 frs	16.5 Gy/3 frs	Not reported	67.31	80.78	CTV + 3 mm	2 yr LC: 86% 3 yr LC: 70%	2 yr OS: 82% 3 yr OS: 54%	Local: 5 Regional: 1 Distant: 1	≥Grade 3: 2 1 feeding tube dependence
Yamazaki H et al. (2014) retrospective/mixed [49]	25	28 (7–128)	35 Gy–70 Gy (Median 50 Gy/25 frs)	12–35 Gy/1–5 frs Median 15 Gy/3 frs	Not reported	Median 68.75	Median 82.5		2 yr LC: 89% 5 yr LC: 71%	2 yr OS: 83% 5 yr OS: 70%	-	≥Grade 3: None
Karam et al., (2014)/retrospective/salivary gland [32]	10	29 (12–120)	Median 64.8, range (50–75.6)	Median17.5, range (10– 30)/3–6 frs	Not reported	87.82 (61.11–113.1)	92.5 (75.91–102.3)	Definitive= GTV + 15– 20 mm Post-op CTV + 10– 20 mm	1-yr LC: 90% 2-yr LC: 80%	1 yr: 100%	Local: 1 Distant: 1	≥Grade 3: None
Kataria et al., (2015)/retrospective/mixed [55]	9	8 (6–19)	54 (50–60)/(25–30)	15 (10–25)/2–5 frs	Median GTV 16.3 (7– 47)	72.7 (62.5–91.2)	87.3 (75–109.5)	GTV +3–5 mm	CR: 55%	Not reported	Distant: 1	≥Grade 3: None
Diaz- Martinez et al., (2018)/retrospective/Sinonasal/ nasopharynx [56]	9	13.3 (4–32)	64.3 (54–70)/(27–35)	13 (12–20)/1 fr	Mean GTV 4.5 (1.17– 8.2)	89.2 (76–120)	107.1 (91.2–144)	Not reported	1-yr LC: 100%	Not reported	Distant: 3	≥Grade 3: None
Baker S et al. (2018)/retrospective/oropharynx Baker S et al. (2019)b retrospective/oropharynx [52]	195	42.8 (2.1–98.6)	46 Gy/23 frs	16.5 Gy/3 frs	Not reported	67.31	80.78	CTV + 3 mm	5 yr LC: 90%	5 yr OS: 66.7%	Local: 18 Regional: 12 Distant: 11	≥Grade 3: 47
Vempati et al., (2020)/prospective/oropharynx [54]	34	50	60–66/30 frs	8–10/1–2 frs	Mean GTVp 70 Mean boost volume 54 (13–185)	72–79.6	86.4–95.5	CTV = GTV + 7 mm PTV = CTV + 3 mm	Median follow up of 50 months LC: 85.3%	Median follow up of 50 months OS: 85.3%	Local: 1 Regional: 2Distant: 4	≥Grade 3: 4 Dysphagia: 1 Pharyngeal hemorrhage: 3

**Table 4 cancers-15-05010-t004:** Salvage SBRT studies for unresectable recurrent or second primary head and neck cancer.

Author (Year)/Design/Subsite	Sample Size (n)	Treatment	rRT Dose (Gy)/Fraction	Radiotherapy Treatment Duration	rRT Tumor Volume (cm^3^), Median (Range)	Median Follow Up (Months)	LC/LRC	Median Survival Rate, Months	Overall Survival Rate, %	Grade 4/5 Late Toxicity, %
Heron et al. (2009)/phase I/Mixed [73]	25	SBRT	25–44 Gy/5 frs	2 weeks	44.8 (4.2–217)		-	6	-	0
Rwigema et al. (2010)/Retrospective/Mixed [72]	85	SBRT	15–44 Gy/1–5 frs	1–38 days	25.1 (2.5–162)	6	1-y LC: 51.2 2-y LC: 30.7	11.5	1-y OS: 48.5 2-y OS: 16.1	0
Heron et al. (2011)/Retrospective/Mixed [74]	70	SBRT +/− cetuximab	20–44 Gy/5 frs	9–14 days	29 (4.8–86.8)	21.3	SBRT alone: 1-y LC: 53.8 2-y LC: 33.6. SBRT + Cetuximab: 1-y LC: 78.6 2-y LC: 49.2	SBRT alone: 14.8 SBRT + Cetuximab: 24.5	SBRT alone: 1-y OS: 52.7 2-y OS: 21.1. SBRT + Cetuximab: 1-y OS: 66 2-y OS: 53.3	0
Comet et al. (2011)/Retrospective/Mixed [69]	40	SBRT +/− cetuximab	36 Gy/6 frs	11–12 days	29.5 (8–85)	25.6	-	13.6	1-y OS: 58 2-y OS: 24	0
Lartigau et al. (2011)/Phase II/Mixed [70]	56	SBRT + cetuximab	36 Gy/6 frs	11–12 days	-	11.4	3 months LC: 91.7	11.8	1-y OS: 47.5	Grade 5:2 patients: (hemorrhage and denutrition)
Cengiz et al. (2011)/Retrospective/Mixed [68]	46	SBRT	18–35 Gy/1–5 frs	Daily	45(3–206)	7	Median PFS: 10.5	1.9	1-y OS: 47	Grade 5:8 patients, 17.8%): carotid blowout
Vargo et al. (2014)/Retrospective/Mixed [17]	132	SBRT + cetuximab	35–40 Gy/5 frs	7–14 days	30.9 (4.4–192.4)	6	1-y LRC: 48	7	1-y OS:38	0
Unger et al. (2010)/Retrospective/Mixed [64]	65	SBRT	21–35 Gy/2–5 frs	Daily	-	16	2-y LRC: 30	12	2-y OS: 41	Grade 4/5 late Toxicity: (6 patients, 9%) arterial bleeding, soft tissue necrosis, fistula formation, and dysphagia requiring hospitalization.
Roh et al. (2009)/Retrospective/Mixed [71]	36	SBRT	18–40 Gy/3–5 frs	Daily	22.6 (0.2–114.9)	17.3	1-y LRFS: 61 2-y LRFS: 52.2	16.2	1-y OS: 52.1 2-y OS: 30.9	Grade 4/5 late Toxicity: (3 patients, 6.8%) (1 bone necrosis, 2 soft tissue necrosis)
[15] et al. (2018)/Retrospective/Mixed	197	SBRT	16–50 Gy/1–8 frs	Every other day	30 (1–427)	24	2-y cumulative LRF: 57	7.8	2-y OS: 16.3	Grade 4/5 late Toxicity: (5% of patients developed carotid blowout syndrome, fistula, and intensive care unit admission)
Ansinelli et al. (2018)/Retrospective/Mixed [75]	45	SBRT	20–42.5 Gy/5 frs	Every other day	34.09 (1.00–258.12)	8.78	1-y LC: 49.6	9.23	1-y OS: 37.7	0

Abbreviations: rRT = re-irradiation, LC = local control, LRC = locoregional control, SBRT = stereotactic body radiotherapy, PFS: Progression free survival, Fr = fraction.

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
