# Peer review of "The Evolving Role of Stereotactic Body Radiation Therapy for Head and Neck Cancer: Where Do We Stand?"

_cancers, 2023, doi:10.3390/cancers15205010_

Round 1

Reviewer 1 Report

Having so immature data, an excursion to radiation biology is recommended. Head and neck SCC have low repair capacity but can proliferate fast. As first line treatment, STx is highly non-standard and should not be performed in locally advanced disease. As a boost treatment, STx should only performed within clinical trials. One should not argue to much with patients who can not tolerate prolonged series of fractionated treatments, as almost all patients who will rofit from radiotherapy can tolerate conventional fractionated or hypofractionated series with or without CTx. 

Author Response

Reviewers' comments:

We are grateful for the dedicated effort you invested in reviewing this review article and for your invaluable insights and recommendations aimed at enhancing its quality.

Reviewer #1: - Comment #1: Having so immature data, an excursion to radiation biology is recommended. Head and neck SCC have low repair capacity but can proliferate fast. As first line treatment, STx is highly non-standard and should not be performed in locally advanced disease. As a boost treatment, STx should only performed within clinical trials. One should not argue to much with patients who can not tolerate prolonged series of fractionated treatments, as almost all patients who will rofit from radiotherapy can tolerate conventional fractionated or hypofractionated series with or without CTx.

Answer #1: We totally agree with reviewer comment. We added a new paragraph in the beginning of the manuscript addressing radiobiological aspects of SBRT. Radiobiological aspects of SBRT High-dose radiation per fraction induces more necroptosis and apoptosis. Consequently, the repair of tumor cells becomes almost impossible, or occurs at an exceedingly low rate, leading to the majority of tumor cells suffering from radiation-induced damage. Moreover, a single high-dose SBRT treatment completely halts the cell cycle at all stages, thereby preventing the redistribution of tumor cells. This high-dose radiation effectively eliminates both oxygenated and hypoxic cells, efficiently eradicating the tumor. In contrast, following conventional radiation therapy, accelerated repopulation of tumor stem cells often occurs after approximately three weeks. However, SBRT treatment is typically completed within one week, effectively spars tumor cells from accelerated repopulation [12-18]. Lines 80-89 Also, we added a simple summary to the manuscript "Currently, Stereotactic Body Radiation Therapy (SBRT) is reserved for head and neck cancer (HNC) patients who are not suitable candidates for conventional radiation therapy and should not be considered as a first line of treatment option and as a boost should be performed in the context of clinical trial. This review aims to explore SBRT's role in different HNC scenarios. It has the potential to greatly impact the clinical practice by providing valuable insights into the appropriate indications for SBRT in HNC treatment, as well as the practical and technical considerations involved in administering SBRT for HNC, SBRT dosage for various HNC scenarios, and treatment results. However, further research is warranted to fully investigate these applications". Lines 34-41

Reviewer 2 Report

The work is well structured and interesting. I think it is an innovative review and deserves publication. In my opinion, typos should be double-checked and corrected, and the tables should be improved.

Quality of English language is adeguate, typos should be double-checked and corrected

Author Response

Response to reviewer comments Reviewers' comments:

We are grateful for the dedicated effort you invested in reviewing this review article and for your invaluable insights and recommendations aimed at enhancing its quality.

Reviewer #2: - The work is well structured and interesting. I think it is an innovative review and deserves publication. In my opinion, typos should be double-checked and corrected, and the tables should be improved

Answer #1: We thank the reviewer for his opinion regarding our work. We double- checked and corrected the typos in the manuscript.

Reviewer 3 Report

Mohamad, et al. present a comprehensive review of SBRT for head and neck cancer.  This literature review is presents a thorough and fair review of the clinical use of SBRT in head and neck cancer patients.  However, this review can be improved by moving the "Practical and technical aspects of SBRT for HNC" to the beginning of the manuscript, as this information is important to better contextualize the rest of the manuscript.  In addition, this section should include information regarding the fundamental difference between SBRT and conventional IMRT.  

Author Response

Response to reviewer comments Reviewers' comments:

We are grateful for the dedicated effort you invested in reviewing this review article and for your invaluable insights and recommendations aimed at enhancing its quality.

Reviewer #3: - Mohamad, et al. present a comprehensive review of SBRT for head and neck cancer. This literature review is presents a thorough and fair review of the clinical use of SBRT in head and neck cancer patients. However, this review can be improved by moving the "Practical and technical aspects of SBRT for HNC" to the beginning of the manuscript, as this information is important to better contextualize the rest of the manuscript. In addition, this section should include information regarding the fundamental difference between SBRT and conventional IMRT.

Answer #3: We thank the reviewer for this comment. We moved the practical and technical aspects of SBRT for HNC to the beginning of the manuscript. We added a paragraph describing the fundamental difference between SBRT and conventional IMRT in the practical and technical aspects of SBRT section lines 91- 98 " Compared with SBRT, Intensity modulated radiation therapy (IMRT) is typically administered over a longer course of treatment, often several weeks, and is better suited for larger or more complex tumors. While both techniques aim to deliver effective radiation therapy with minimal damage to healthy tissue, SBRT's emphasis on precision, accuracy, rapid treatment, meticulous target volume delineation, no or minimal clinical target volume (CTV), possibly tighter planning target volume (PTV) margin, steep dose gradient, and larger dose per fraction which make it particularly well-suited for certain clinical scenarios [11, 15]."

Round 2

Reviewer 1 Report

The revised version has been greatly improved and this reviewer fully agrees with the conclusions. However, he does not fully agree with the radiobiological excurses. Important for the choice of the fractionation scheme are differences in critical radiobiological characteristics between normal tissues within the high dose volume and the tumor. Head and neck SCC have low repair capacities or fractionation sensitivities where hyperfractionation works. A lot of normal tissues have higher repair capacities, so it should be possible with stereotactic radiotherapy to keep those normal tissues out of the high dose volume, e.g. vessels or major nerves. In addition, normal mucosa has a very high repopulation capacity that cannot protect mucosa during SBRT, so it must be able to spare larger mucosal surfaces form the high dose region, and this makes smaller volumes mandatory. If the tumor is hypoxic you waste dose to the hypoxic region that could reoxygenate during more fractionated regimens. Therefore, the hypoxic region should be limited, as in smaller recurrences. Please comment on.

Author Response

: Thank you for your valuable comment. We added a new paragraph to radiobiological principles of SBRT for HNC